# Molecular characterization of hepatitis B virus genotypes A and D among inmates and blood donors in Northeastern Kenya

**Vincent Bahati Odallo**[1], **Okoti P. Aluora**[1,2], **Wallace Bulimo**[3], **George Gachara**[1*]

**1** Department of Medical Laboratory Science, Kenyatta University, Nairobi, Kenya, **2** Department of Health Sciences, The Nairobi National Polytechnic, Nairobi, Kenya, **3** Centre for Virus Research, Kenya Medical Research Institute (KEMRI), Nairobi, Kenya

* gachara.george@ku.ac.ke

## Abstract

### Introduction

Hepatitis B virus (HBV) persists as a major global public health burden, with hyperendemic prevalence in sub-Saharan Africa. Populations with elevated exposure to percutaneous transmission risks, including incarcerated individuals and healthcare workers demonstrate heightened HBV susceptibility. Despite this, genomic data from Northeastern Kenya and Kenyan prison populations remain scarce.

### Objective

To characterize HBV genotypic diversity circulating in Northeastern Kenya, among low-risk (blood donors) and high-risk (prison inmates) populations.

### Methods

A cross-sectional investigation compared HBV seroprevalence and genotypes between incarcerated individuals (n = 130) and voluntary blood donors (n = 130) in Garissa County, Northeastern Kenya. Serum samples were subjected to Hepatitis B surface antigen (HBsAg) screening, PCR amplification of a 940-bp overlapping surface/polymerase gene, sequencing, and phylogenetic/recombination analyses of the resulting sequences.

### Results

Seroprevalence was higher among incarcerated individuals (5.4%, 7/130) than blood donors (3.1%, 4/130). Hepatitis B Virus DNA was detected in 22 samples which were all successfully sequenced. Genotype D dominated both cohorts (81.8%), while genotype A sub genotype A1 occurred exclusively in incarcerated participants (18.2%). All genotype D strains were recombinants: D/A (61%) and D/E (39%). Sequences are accessible in GenBank (accession numbers: PV816552–PV816573).

**Data availability statement:** The Sequences from the study are deposited in GenBank under Accession numbers: PV816552-PV816573.

**Funding:** The author(s) received no specific funding for this work.

**Competing interests:** The authors have declared that no competing interests exist.

## Conclusions

This first genomic study of HBV in Kenyan prisons confirms incarcerated populations as high-risk. The predominance of genotype D—an unusual finding in this region and high recombinant frequency (100% of genotype D strains) underscore significant viral evolution. Expanded genomic surveillance is imperative to define HBV diversity, inform vaccine efficacy monitoring, guide screening policies and optimize control strategies in Northeastern Kenya.

## Introduction

Hepatitis B virus (HBV) remains a major global health burden, with an estimated 254 million individuals living with chronic infection in 2022 and an estimated 1.1 million annual deaths primarily due to liver cirrhosis and hepatocellular carcinoma [1]. The virus is genetically diverse and is classified into ten genotypes (A–J) with distinct geographic distributions and clinical implications [2–5]. Sub-Saharan Africa, including Kenya, has been identified as a region with a high HBV burden, predominantly characterized by genotypes A, D, and E [6–9]. These genotypes have shown a propensity for intergenotypic recombination, impacting the virus's evolution, enhancing its adaptability, transmission, and persistence within populations [10–12].

HBV recombination is not merely a molecular curiosity but has significant implications for public health. Recombination events can alter viral fitness, impact the course of disease, and create challenges in diagnosis, treatment, and prevention. Previous studies in Africa have reported the circulation of HBV recombinants involving genotypes A, D, and E, suggesting that these strains are not only evolving but also influencing regional epidemiology [11]. In Kenya, earlier research has identified recombinant strains, such as D/E recombinants, but comprehensive molecular data on recombination patterns and their clinical consequences remain limited [8].

Makhoha et al., (2023) published a systematic review and meta-analysis of the estimates of HBV infection in Kenya and obtained an overall pooled prevalence estimate of 7.8% [13]. Available studies have traditionally surveyed the general Kenyan population, as well as specific populations such as blood donors, HIV patients, pregnant women and healthcare workers. Geographically, studies have ignored the northern and northeastern regions of the country [14] where anecdotal reports suggest a higher HBV burden. Additionally, current studies have ignored the incarcerated population which is a known HBV high-risk group. Consequently, the exact picture of the HBV burden in the country is not fully understood and may be underestimated.

HBV genotypes influence disease progression, response to treatment and likelihood of mutations conferring vaccine escape or antiviral resistance [15]. Genotype D has been associated with more severe liver disease in some populations. However, its epidemiological and clinical implications in Kenya remain poorly defined. This study sought to address these gaps by conducting a molecular analysis of HBV in a group of inmates and voluntary blood donors in the northeastern part of Kenya.

Prison inmates are often at higher risk of HBV infection due to shared living spaces, limited access to healthcare, and behaviours such as drug use or unsafe tattooing practices. Conversely, voluntary blood donors represent a low-risk population routinely screened for infectious diseases, offering valuable insights into HBV strains circulating within the general population. The primary aim of the study was to characterize the burden and genotypic diversity of HBV strains circulating within this neglected population. The findings are expected to provide clarity on the true HBV burden in the country in an unstudied region and population in the country, which is important in developing targeted interventions to curb HBV transmission and progression in line with the global agenda to eliminate viral hepatitis by 2030.

## Materials and methods

### Study setting

The study was conducted in Garissa County in two study sites namely Garissa County Referral Hospital (GCRH) and Garissa Main Prison within the Township area. This County is found to the Northeastern part of Kenya. GCRH receives patients from Garissa County and also neighbouring counties such as Kitui, Tana River and Wajir. Equally, Garissa Main Prison is a host to in-mates from both Garissa County and other parts of the country.

### Study population

This involved two groups; voluntary healthy blood donors and prison inmates. The blood donors presented at GCRH while the inmates were incarcerated at Garissa Main Prison. Irrespective of gender and other socio-economic characteristics, all voluntary blood donors were selected according to the Kenya National Blood Transfusion Services (KNBTS) criteria. The study included blood donors aged 18–65 years and weighing at least 50 kg, as well as inmates aged 18 years and above.

A sample size of 130 for each of the study population was determined based on an estimated HBV prevalence of 5% in high-risk populations (prison inmates) [16] and 3% among low-risk populations (blood donors) [17]. This was calculated using a standard sample size formula for comparing two proportions [18] powered at 80% ($\alpha = 0.05$) to detect a $\geq 2$-fold seroprevalence difference between cohorts.

### Ethical considerations

Ethical approval was sought and granted from Kenyatta University Ethics and Review Committee (KU-ERC, application number PKU/2043/I1190). A research permit from the National Commission for Science, Technology & Innovation (NACOSTI) to conduct the research was obtained under License No. NACOSTI/P/20/4150. To further facilitate the study, clearance was sought and obtained from the Kenya National Blood Transfusion Services (KNBTS) trainings board and the Commissioner General and the Trainings Board of the Kenya Prisons Service. Each of the study participants gave informed consent after the study protocol was explained to them in writing and participated without coercion or remuneration.

### Sample collection and processing

Sample collection for the inmates was done at Garissa Main Prison dispensary located within the prison compound while for the blood donors, it was done at GCRH Blood Transfusion Unit (BTU) between 03/08/2020 and 31/12/2020. Approximately 4 ml of blood was collected from a selected vein of each inmate and an equivalent amount was tapped from the blood bag of each participating blood donor and then placed into a plain vacutainer. Serum for serological work was prepared by centrifuging the blood at 3000xg for five minutes. Serum preparation and serological testing was done on site at Garissa Main prison dispensary laboratory for the inmates and at GCRH laboratory for the blood donors. Serum was aseptically aspirated and dispensed into already prepared cryotubes, packaged in a cooler box and transported to Kenyatta University (KU) laboratories where they were refrigerated at −80°C awaiting further processing.

## Serological testing

All the serum samples were screened for the hepatitis B virus surface antigen (HBsAg) using a rapid immunochromatographic test cassette (Amitech Diagnostics Inc.) following manufacturer's instructions.

## DNA extraction

The HBV DNA was extracted using the Isolate II Genomic DNA Extraction kit (Bioline, Germany) according to manufacturer's instructions. The eluted DNA was then stored at −20°C awaiting amplification process.

## PCR and sequencing

The overlapping region of the HBV P and S genes was amplified using a heminested PCR protocol. The PCR amplification reaction was performed in a final volume of 25 µl using universal sets of primers and protocols described previously by Chook et al., (2015) [19] to amplify the region 251–1190 from the EcoR1 site of the viral genome.

The first round of PCR reaction contained 0.5µl My Taq DNA polymerase (Bioline, Meridian Life Science, Memphis, USA), 10 µl 5X My Taq reaction buffer, 0.5µl each of 20µM first round primers (251f and 1797r), 7.5µl of nuclease free water and 6µl of the DNA template. Both initial denaturation and denaturation were performed at 95°C for all the samples. An initial denaturation for 2 minutes was followed by 35 cycles of denaturation for 15s, annealing at 58°C for 30s and extension at 72 °C for 30s. A final extension was performed at 72 °C for 2 mins while a final hold was set at 4 °C to terminate the reaction. The second round PCR reaction was performed using 5 µl of the first round PCR reaction product as template while using the second set of primers (251f and 1190r) under the same reaction conditions to amplify a 940 bp amplicon. Each run utilized negative and positive controls.

The PCR products were visualized and resolved in a 1.5% agarose gel electrophoresis stained with 5µl ethidium bromide (EtBr) and examined using a UV transilluminator. Purification of the HBV PCR positive amplicons before sequencing was done by treating with ExoSAP-IT™ (ThermoFisher Scientific, CA, USA). The purified PCR products were then sequenced at an outsourced facility (Inqaba Biotec, Pretoria, South Africa) using bidirectional Sanger sequencing on an ABI 3500xL (Applied Biosystems, CA, USA) sequencer.

## Data analysis

All data was entered onto a Microsoft Excel°, 2021 and descriptive analysis conducted using IBM SPSS Statistics (version 20) [20]. The resulting forward and reverse sequences were assembled into contiguous nucleotide sequences and manually edited in BioEdit version 7.25 [21] sequence editor to remove primer sequences. A quality control check of the sequences for contamination was conducted using the online basic local alignment search tool (BLAST) [22]. Clustal W program implemented in BioEdit [21] was used in alignment of the resulting nucleotide sequences. The geno2Pheno database (https://hbv.geno2pheno.org) and phylogenetic trees were used for HBV genotypic determination. MrBayes program version 3.1.2 was used for phylogenetic reconstruction [23]. First, genotype specific sequences were downloaded from Genbank and the resulting fasta file which included the study sequences was converted into the nexus format using Concatenator (http://cobig2.fc.ul.pt.). The nexus file was then run for 1 million generations sampling at every 100th generation with a burn in setting of 10% of generations. The GTR + G model (general time-reversible model with gamma distributed rates of variation among sites) was used. Visualization of the resulting phylogenetic tree and determination of the HBV genotypes was done using Fig Tree version 1.3.1 (http://tree.bio.ed.ac.uk/software/figtree/).

Recombination was analysed for sequences whose sub-genotype identity could not be resolved by phylogenetic reconstruction using the HBV NCBI genotyping tool (https://www.ncbi.nlm.nih.gov/projects/genotyping/formpage.cgi).

## Results

### Seroprevalence of HBV

A total of 260 study participants were enrolled in the current study, out of whom 130 were voluntary blood donors and 130 were inmates. Among the blood donors, 4/130 (3.1%) were HBsAg seropositive while among the inmates, 7/130 (5.4%) were HBsAg seropositive as shown in Table 1. Although seroprevalence was higher among inmates (5.4%) than blood donors (3.1%), this difference was not statistically significant. (p = 0.31, $\chi^2 = 0.366$)

### Prevalent HBV genotypes

In a total of twenty-nine (29) samples, a 940 bp HBV DNA was successfully amplified using PCR. Out of these, twenty-two (22) were successfully sequenced majority (18/22) of whom were HBV seronegative (Gene bank accession numbers: PV816552-PV816573). This indicates a high prevalence of occult HBV infection (OBI). The geno2Pheno database showed that out of the 22 HBV sequences, 18 (81.8%) were identified as genotype D sub-genotype D4 among both blood donor 7/18 (38.9%) and inmate 11/8 (61.1%) populations. Genotype A sub genotype A1 was identified in 4 (18.2%) sequences which were all derived from inmates as shown in Table 2 below.

The phylogenetic analysis confirmed the identities of the circulating HBV genotypes and also the A1 sub genotype. However, the sub-genotype identity of the identified genotype D previously identified as D4 by the geno2pheno database could not be resolved phylogenetically. The genotype D sequences from this study did not cluster with published sub genotype D4 sequences but instead distinctly clustered away from the other sub-genotypes. This cluster of local genotype D sequences from the study was well supported with a posterior probability of 91% as shown in Fig 1. The clustering of all the genotypes and sub-genotypes was well supported with posterior probabilities >80%.

### HBV genotype D recombinants

The 18 genotype D sequences were checked for recombination in an attempt to resolve their sub-genotype identity. All the sequences were found to be recombinants with genotype A and E. There were 11/18 (61%) genotype D/A recombinants and 7/18 (39%) genotype D/E recombinants as shown in supplementary S1 and S2 Figs. The recombination breakpoint was observed between position 501 and 800 in all the recombinants.

Table 1. HBV seroprevalence among blood donors and inmates.

| HBV Seropositivity | Blood Donors | | Inmates | |
|---|---|---|---|---|
| | Frequency | Percent (%) | Frequency | Percent (%) |
| Negative | 126 | 96.9 | 123 | 94.6 |
| Positive | 4 | 3.1 | 7 | 5.4 |
| Total | 130 | 100.0 | 130 | 100.0 |

Table 2. HBV Genotypes among the study participants.

| Genotype | Study Population | No. of sequences | % of sequences |
|---|---|---|---|
| D | Inmate [11] Donor [7] | 18 | 81.8 |
| A | Inmate | 4 | 18.2 |
| Total | | 22 | 100.0 |

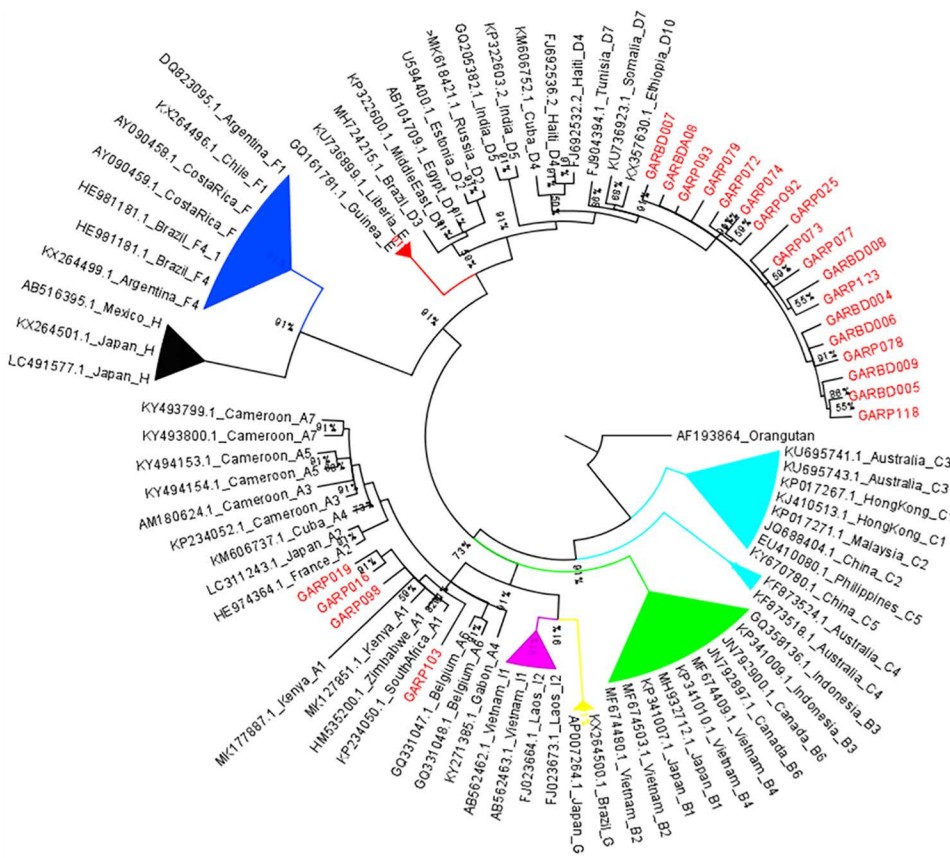

**Fig 1. Phylogenetic tree of HBV sequences from this study alongside published sequences.** A bayesian rooted phylogenetic tree constructed using MrBayes ver 3.1.2 of the HBV sequences from this study labeled in red at the taxa. Sixty-seven (67) global sequences obtained from GenBank were included to support tree topology and genotype identification. They are labeled with their gene bank accession numbers, country of origin and sub genotype. The study sequences from this study are labeled GARD and GARP to indicate sequences from blood donors and prison inmates respectively.

## Discussion

Our study findings reported herein contribute to the 2030 Agenda for Sustainable Development [24] specifically target 3.3 by identifying the burden of HBV in a previously unstudied region and risk group. In Kenya, there is scanty information on hepatitis infection among prison inmates and the populations in the Northern region [14]. This information is important for developing targeted prevention and treatment interventions.

The seroprevalence of HBV in the two study populations was observed to differ. Inmates who are classified as high risk for HBV infection had a HBsAg seropositivity of 5.4% compared to 3.1% among the blood donors who are classified as low risk. These findings are in agreement with a recent systematic review of HBV prevalence in Kenya using 23 studies published between January 1990 and December 2021 [13] that found a pooled HBV prevalence of 3.1% (95% CI 2.62–4.01%) among the low-risk group which includes blood donors. It also found a pooled prevalence of 5.58% (95% CI 3.46–7.7% and, 6.17% (95% CI 4.4–9.94) among moderate risk and high-risk groups, respectively [14]. A number of factors have been identified as drivers of HBV spread in prisons. These include congested living conditions, inadequate healthcare facilities, limited access to preventive measures, unprotected sexual intercourse, injection drug use (IDU), tattooing and other forms of skin piercing [25,26]. The current study therefore affirms the categorization of blood donors as

a low-risk group and inmates as a high-risk group in HBV transmission. It is important to note that the systematic review observed that no study had been conducted in Northern Kenya and also among inmates in the country.

The genotypic analysis performed in this study using the Geno2Pheno database identified genotype D, sub-genotype D4 as the dominant HBV genotype, accounting for 81.8% of the sequenced samples across both blood donor and inmate populations. Among the inmate population, a smaller proportion of subjects exhibited genotype A, sub-genotype A1, underscoring the coexistence of multiple genotypes within this cohort. The exclusive detection of sub-genotype A1 among inmates, suggests a possible distinct transmission dynamics within the incarcerated population. Earlier studies in Kenya have documented the circulation of HBV genotypes A, D, and E [7,27,28], with sub genotype A1 consistently identified as the predominant one [6,27]. However, the dominance of genotype D, as observed in this study, represents an unusual trend, diverging from the established pattern in previous studies. The genotype was observed in 61.1% of the inmate HBV DNA positive samples and in all the HBV DNA positive blood donor samples. Genotype D dominance, as observed here, may impact disease progression and treatment outcomes given evidence of higher risk of liver disease progression and escape mutations associated with certain genotype D sub-genotypes [29].

The distribution of HBV genotypes and sub-genotypes is largely influenced by ethnicity and migration [30]. Consequently, the predominance of genotype D in Northeastern Kenya, as reported here, could be attributed to the socio-demographic dynamics of the region. This area is predominantly inhabited by the Somali ethnic group, which constitutes a significant proportion of the population in the Horn of Africa. Studies have suggested that HBV genotype D is widespread in East Africa, including among Somali populations [31,32]. The potential introduction of genotype D into Northeastern Kenya through migration or trade with neighboring regions may have further contributed to its prevalence in this region.

The genotype D sequences in this study revealed unexpected phylogenetic patterns. Despite utilizing published genotype D sub-genotype reference sequences, the local genotype D sequences failed to align with any recognized sub-genotype (D1-D10). Instead, these sequences formed a separate, well-supported cluster (posterior probability = 91%), distinct from established sub-genotypes. The absence of clustering with recognized HBV genotype D sub-genotypes raises intriguing possibilities. One explanation is that the observed sequences represent a new HBV sub-genotype within genotype D, defined by a genome divergence of at least 4%, as established in HBV classification guidelines [31,33]. We could not test this hypothesis since the study did not sequence full genomes. Alternatively, these sequences could represent genotype D recombinants, as recombination events are not uncommon in HBV and have been documented in other studies [11,12]. This hypothesis is further supported by the high genetic diversity within genotype D, as well as reports of recombinant strains globally, including in Africa [8,11,34].

Previous studies in Kenya have detected both genotype D sub-genotypes D1 and D4 [27,28]. However, outliers that do not conform to any recognized sub-genotype have also been reported, such as an unclassified HBV genotype detected in 2013 [27]. The presence of D/A and D/E recombinants, is not totally unexpected considering that genotypes A, D and E have been the most frequently detected in Kenya [35]. While no report of the recombination breakpoint identified in this study is available, D/E recombinants have previously been reported in Kenya [27], Uganda [36], Sudan [32,37], Niger [38] and Ghana [39]. The D/A recombinants have also been reported in Ghana and Guinea [40].

The high frequency of HBV D/A and D/E recombinant strains observed in this study (81.8%) underscores the significant genetic plasticity of HBV in Northeastern Kenya. Such recombination between co-circulating genotypes is a recognized evolutionary phenomenon [41] and complicates molecular surveillance by obscuring true transmission networks. Critically, these recombinant forms may impact diagnostic accuracy if commercial assays target recombinant-prone genomic regions and could theoretically alter vaccine efficacy or antiviral response, though clinical implications of D/A recombinants remain uncharacterized and warrant further investigation.

Our findings must be interpreted in light of two limitations. First, HBsAg screening relied on rapid immunochromatographic tests (sensitivity: 90–95%) rather than higher-sensitivity EIAs (>99%) or PCR [1], potentially underestimating true seroprevalence. Second, sequencing of only a 940-bp P/S fragment precluded comprehensive recombination mapping

(e.g., breakpoint identification), full sub-genotype characterization, and assessment of drug-resistance mutations. Thus, whole-genome sequencing of HBV-positive samples from this region is essential to resolve recombinant structures, detect emerging variants, and clarify clinical implications.

Notwithstanding its limitations, this study provides the first genomic characterization of hepatitis B virus (HBV) in Kenya's Northeastern region, revealing two critical findings: a dominant circulation of genotype D among incarcerated populations (representing 81.8% of HBV-DNA positive samples) and blood donors, coupled with the emergence of recombinant strains (D/A: 61%; D/E: 39%) indicative of novel viral variants. Though there is little difference in nucleos(t)ide analog drug resistance between genotype A and D, genotype D often has a poorer IFN-α response than genotype A. Genotype D also has been linked to more severe disease and hepatocellular carcinoma. These results significantly advance our understanding of HBV's evolutionary trajectory in East Africa, a region where genotype A has historically predominated. The high recombinant frequency signals active viral adaptation with tangible public health implications: diagnostic reliability may be compromised if commercial assays target recombinant-prone genomic regions, while vaccine efficacy could theoretically be undermined through immune-escape mutations, and antiviral treatment efficacy may require reassessment should recombinants alter drug-resistance profiles. To address these challenges, we recommend implementing regionally tailored genomic surveillance programs, validating existing diagnostics against circulating recombinant strains, and prioritizing whole-genome sequencing to resolve sub-genotypic diversity and recombination breakpoints. This calls for the government in collaboration with global partners to invest in infrastructure and review policy to mainstream HBV whole genome surveillance in order to optimize HBV prevention and treatment strategies in Kenya.

## Supporting information

**S1 Fig. Figure of a recombination analysis of a HBV genotype D sequence showing D/E recombination.** (TIF)

**S2 Fig. Figure of a recombination analysis of a HBV genotype D sequence showing D/A recombination.** (TIF)

## Acknowledgments

The authors would like to extend their sincere gratitude to the study participants for their invaluable contribution and consent to participate in this research. Without their cooperation, this study would not have been possible. We also acknowledge the support of the healthcare staff and institutions involved in data collection and sample processing. Special thanks are given to the laboratory technicians for their meticulous work in sequencing and analyzing the samples.

## Author contributions

**Conceptualization:** George Gachara, Vincent Bahati Odallo, Okoti P. Aluora.

**Data curation:** Vincent Bahati Odallo.

**Formal analysis:** George Gachara, Vincent Bahati Odallo.

**Investigation:** George Gachara, Vincent Bahati Odallo.

**Methodology:** George Gachara, Vincent Bahati Odallo, Okoti P. Aluora, Wallace Bulimo.

**Project administration:** Wallace Bulimo.

**Supervision:** George Gachara, Okoti P. Aluora, Wallace Bulimo.

**Writing – original draft:** Vincent Bahati Odallo.

**Writing – review & editing:** George Gachara, Okoti P. Aluora, Wallace Bulimo.

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
