## [Decision Letter · Decision Letter 0]

18 Nov 2025

Dear Dr. Gachara,

Thank you for submitting your manuscript to PLOS ONE. After careful consideration, we feel that it has merit but does not fully meet PLOS ONE’s publication criteria as it currently stands. Therefore, we invite you to submit a revised version of the manuscript that addresses the points raised during the review process.

Thank you for submitting your manuscript. The study addresses an important topic; however, the Methods section requires substantial revision to ensure clarity, reproducibility.

Please address all the comments suggested by the reviewers. Also ensure you proofread the manuscript for grammatical error and consistency with labelling reagents, instruments and analysis tools.

We look forward to receiving your revised manuscript.

Kind regards,

Maemu Petronella Gededzha, Ph.D

Academic Editor

PLOS ONE

Journal Requirements:

Additional Editor Comments (if provided):

Please address all the comments from the reviewers and on the attached document.

Reviewers' comments:

Reviewer's Responses to Questions

**Comments to the Author**

1. Is the manuscript technically sound, and do the data support the conclusions?

Reviewer #1: Yes

Reviewer #2: Partly

Reviewer #3: Yes

2. Has the statistical analysis been performed appropriately and rigorously?

Reviewer #1: No

Reviewer #2: N/A

Reviewer #3: No

3. Have the authors made all data underlying the findings in their manuscript fully available?

Reviewer #1: Yes

Reviewer #2: Yes

Reviewer #3: Yes

4. Is the manuscript presented in an intelligible fashion and written in standard English?

Reviewer #1: Yes

Reviewer #2: No

Reviewer #3: Yes

Reviewer #1: A) GENERAL COMMENTS:

This manuscript presents a well-conceived and scientifically relevant investigation into the seroprevalence and genotypic diversity of Hepatitis B virus (HBV) among two distinct populations in Northeastern Kenya prison inmates (high-risk) and voluntary blood donors (low risk). The study fills a significant knowledge gap, as molecular data from both Kenya’s northeastern region and incarcerated populations have been historically lacking. The work is timely and aligns with global health priorities, including WHO’s 2030 hepatitis elimination targets (SDG 3.3).

B) ABSTRACT

The abstract is scientifically strong, data-rich, and well aligned with the conventions of high-impact infectious disease or molecular epidemiology journals. It has a clear structure, and the key findings are compelling and novel for the region. The authors should proofread the abstract; it seems AI was used to generate the abstract, hence there are many typos which need to be worked on.

C) INTRODUCTION

The introduction is scientifically sound, well-structured, and shows a solid grasp of HBV molecular epidemiology and public health context.

Minor comments:

• Some sentences could be more concise, for example, “These genotypes have shown a propensity for intergenotypic recombination…” could be tightened for flow.

• The citation style (e.g., “Makhoha et al., 2023 recently published...”) should be in past tense without “recently,” as that becomes outdated; also, remove first names in scientific writing.

• The paragraph on study populations (inmates vs. donors) is informative but slightly repetitive, it reiterates content from the methods and could be condensed.

• The final sentence (“in line with the global agenda…”) is effective but would benefit from a stronger closing statement emphasizing the novelty or expected contribution of this study.

D) METHODS

• The method is well structured and fairly comprehensive for a virology paper describing HBV genotyping among human subjects in Kenya. However, from a reviewer’s perspective, there are both strong scientific elements and a few weaknesses that could be improved to make it read more professionally and to ensure reproducibility and clarity.

• The authors clearly stated the ethics approval and included the inclusion of the consent procedure, which satisfies the international standards. This section demonstrates procedural transparency and biosafety awareness.

• The method sections aligns well with accepted HBV genotyping protocols, particularly the heminested PCR spanning P/S overlap (EcoR1 site region).

Comments:

1. Writing and organization

• Some sentences are too long or contain redundant information, e.g., lines 135–139 could be shortened or merged for clarity.

• There are minor grammatical inconsistencies (“then sequences” “then sequenced”; “was then sequences by bidirectional…” “was then sequenced bidirectionally”).

Suggestion: Edit for flow and consistency (use past tense uniformly, e.g., “was performed,” “was analyzed”).

2. Scientific details missing or vague

• Primer sequences are not explicitly listed; the reference “(17)” should be clearly identified and verified. One would expect at least the forward and reverse primer sequences or accession numbers.

• The annealing temperatures and cycle numbers are provided, but the amplicon size (bp) is not stated.

• The sequencing service could specify location or facility (e.g., “sequenced at Kenyatta University molecular lab or outsourced to …”).

• It is unclear how quality control of sequences (trimming, contamination check) was performed before alignment.

Suggestion: Add brief information on amplicon size, primer details, and QC filtering steps.

• Although the use of MrBayes is correct, the model of nucleotide substitution (e.g., GTR+G+I) used in the Bayesian analysis is not mentioned.

• Tree validation metrics (posterior probability thresholds, burn-in value, number of independent runs) are missing.

• There is no mention of how mixed genotypes or ambiguous bases were handled.

Suggestion: Include these parameters to strengthen reproducibility and perceived analytical rigor.

The overall tone is informative, but the phrasing could be made more formal and concise.

Example revision:

“Approximately 4 mL of venous blood was collected into plain vacutainers from each participant”

is preferred over

“Approximately 4 ml of blood was collected from a selected vein of each individual into a plain vacutainer.”

E) RESULTS SECTION

The Results section presents key findings on HBV seroprevalence and genotypic diversity among two defined populations (blood donors and inmates) and is overall scientifically credible and well structured, though it could be strengthened in presentation, statistical detail, and interpretative depth.

General Impression

The Results section is clear, methodologically consistent with the described procedures, and organized in a logical order (seroprevalence, genotyping, phylogenetic confirmation, recombination). The authors effectively summarize their findings with quantitative data, appropriate referencing of figures and tables, and a concise narrative. However, while the data presentation is generally sound, the section would benefit from greater statistical rigor, contextual interpretation, and improved figure/table integration.

Comments:

1. Seroprevalence of HBV

The authors clearly state the total sample size (n=260), balanced between groups (130 each), which facilitates comparability. Seroprevalence rates (3.1% and 5.4%) are reported precisely and supported by a chi-square test (p=0.31), demonstrating the authors attempted a statistical evaluation. The distinction between inmates and blood donors is clearly maintained, providing social epidemiological context.

• The confidence interval (CI) or exact chi-square value, which would improve statistical transparency is not defined or included. The phrase “did not reach statistical significance (p=0.31)” is correct but should be contextualised through discussing sample size limitations or low statistical power. Table 1 is mentioned but not fully integrated into the text (no data summary within the text).

• The Results section omits any age, sex, or risk factor stratification, which could enrich epidemiological interpretation.

• The seroprevalence analysis is scientifically sound but statistically underdeveloped and somewhat descriptive. Including more analytical insight (e.g., odds ratios or subgroup trends) would strengthen the epidemiological message.

2. HBV DNA Detection and Genotyping( Method must be self-descriptive) HBV DNA and Genotyping of what?

• The author reported successful amplification from 22 samples is clear, and inclusion of GenBank accession numbers (PV816552–PV816573) supports data transparency and reproducibility.

• Genotype proportions (D = 81.8%, A1 = 18.2%) are numerically clear and properly linked to population origin.

• The use of the geno2Pheno database is appropriate and methodologically sound.

• The authors report that “several seronegative samples” were successfully amplified this intriguing observation is scientifically important but underexplained. It suggests occult HBV infection (OBI), which warrants at least a brief acknowledgement even in the Results section.

• The genotype distribution table is referred to as Table 1, but it should be Table 2 to maintain consistency (Table 1 has already used for seroprevalence).

• The authors might clarify how the 22 samples were selected; were these all HBsAg-positive, or did they include a random subset? This is key for interpretation.

3. Phylogenetic Analysis

• The phylogenetic results are well described: genotype confirmation, sub-genotype verification (A1), and Bayesian posterior probability (91%) show appropriate analysis depth.

• The authors mentioned non-clustering with D4 references adds originality to the findings, suggesting a potentially distinct or novel D variant in the local population.

• The latter probability (91%) is acceptable but could be accompanied by a mention of tree topology metrics or branch support for other genotypes for completeness. The quantitative descriptors of the sequence divergence or clustering distance are not defined

• The statement “distinctly clustered away from other sub-genotypes” should be supported by describing whether this divergence indicates a new lineage, regional adaptation, or recombination.

• The recombination results are briefly stated without genomic detail (e.g., breakpoint positions, affected regions).

• The analysis would be stronger if the authors indicated the potential epidemiological or clinical implications of these recombinants, e.g., whether such patterns are consistent with previous East African HBV diversity.

In order for the paper to meet the standards, the author should work on the following

• Inclusion of confidence intervals, chi-square values, and possible multivariate associations.

• Improved linkage between figures/tables and text.

• Brief but data-driven interpretation of notable findings (e.g., recombinants, seronegative amplifications).

• Minor stylistic editing for precision and consistency in table/figure numbering.

F ) DISCUSSION

• This is an excellent and scientifically mature Discussion section that demonstrates a solid understanding of both global HBV control goals and regional molecular epidemiology.

• The authors successfully link their findings to global health targets, national data gaps, and molecular-level implications. However, as a high-level reviewer, I would describe it as analytically strong but slightly overloaded, with some redundant narrative and areas where interpretive depth or flow could be refined to reach a top-tier publication standard.

Comments:

Overall Impression

The Discussion is scientifically sound, contextually rich, and globally aligned, addressing public health, molecular virology, and evolutionary aspects of HBV. It communicates significance effectively and situates findings within both Kenya’s epidemiological landscape and the broader African genomic context. The authors clearly understand how their results contribute new regional insight, especially regarding the novel predominance of genotype D and high recombinant diversity (D/A and D/E).

The authors open by linking their study to the UN SDG 3.3 and WHO’s 2030 hepatitis elimination goals, a globally relevant framing that gives the discussion strong policy alignment. They effectively justify the study’s focus on underrepresented populations (inmates, Northern Kenya), demonstrating clear research need.

• However, the paragraph reads slightly like an introduction rather than a discussion. While context is valuable, a more analytical transition (e.g., “Our findings contribute to this goal by…”) would make it flow better into interpretation. The Citations to WHO and SDG literature could be condensed to avoid redundancy.

• The contextual introduction is excellent in scope but could be streamlined for brevity and better transition into study interpretation.

In lines 238-249, the author interpreted the seroprevalence findings and skillfully related their results to a recent national systematic review, quantitatively comparing prevalence values and confirming known risk stratification (blood donors as low-risk, inmates as high-risk). The discussion acknowledges the knowledge gap in Northern Kenya and among inmates clearly highlighting novelty and relevance.

The following should be worked on :

• There is little exploration of potential behavioural or environmental factors driving higher inmate prevalence (e.g., overcrowding, shared instruments, limited healthcare access).

• The text is descriptive and could integrate a short epidemiological interpretation beyond numeric comparison.

• The discussion is well-documented and comparative, but under-discussed regarding underlying causes and implications for screening policy.

• Excellent reporting and interpretation of genotypic findings, clear recognition that genotype D dominance is unexpected for Kenya, where A1 typically predominates.

The linkage of genotype distribution to ethnic and migratory factors in the Somali-dominated Northeastern region demonstrates sophisticated epidemiological reasoning and familiarity with regional molecular diversity.

• The text could make clearer whether genotype D dominance was observed across both populations or more pronounced in a specific one (this distinction is important for epidemiologic inference).

• The discussion would benefit from hypothesising about possible sources or routes of introduction, for instance, historical or cross-border movement from Somalia or Ethiopia.

• A sentence or two connecting genotype D’s clinical implications (e.g., treatment response to tenofovir or interferon) would enhance translational impact.

• This section is scientifically rigorous and insightful but could emphasise why genotype D matters biologically beyond its novelty.

• The recombination discussion (lines 290–310) becomes somewhat repetitive, restating similar concepts (migration, cross-border mixing, recombination potential). It could be condensed without losing meaning.

• The authors could explicitly indicate whether their D/A and D/E recombinants share breakpoint regions previously reported in Africa; this would strengthen the evolutionary argument.

• The text could better differentiate between what was observed (data-driven) and what is hypothesised (interpretive).

Molecular interpretation is very strong, and could benefits from a tighter focus and distinction between confirmed and speculative findings

• (Lines 316–340), Public Health and Diagnostic Implications were explicitly discussed; however, the potential impact on therapeutic outcomes could be supported by referencing known genotype D-associated resistance patterns (e.g., lower interferon response).

The final recommendations could briefly mention capacity-building needs (infrastructure, partnerships) for genomic surveillance in Kenya.

Overall, the discussion section closes the paper strongly. It clearly articulates relevance and next steps, though a final synthesis paragraph summarizing the key contributions. The paper would read well if the authors, if the authors condensed repetitive paragraphs (Especially those on recombination and migration. The authors should provide more mechanistic and clinical context for genotype D implications, and the concluding paragraph should explicitly state the contributions and next steps.

Reviewer #2: Title:

-The mention of genotype D in the title is ambiguous, as there are also reports on genotype A. Please clarify whether genotype A samples were excluded from the analysis or adjust the title accordingly.

Manuscript Text:

-Review the manuscript for grammar, sentence structure, and scientific writing style.

-Correct issues related to sentence case, punctuation (commas, full stops, etc.), and overall readability.

-Revise unclear or awkward sentences as previously suggested.

-Remove redundant or misplaced information throughout the text to improve flow and conciseness.

-Include missing details on statistical analysis, including the frequency and distribution of results.

-Avoid repetition of information, especially in the results and discussion sections.

-Ensure all factual statements are properly referenced where applicable.

-Add missing details on reagents and materials, including manufacturer and source (e.g., ExoSAP-IT™ (ThermoFisher Scientific, CA, USA)).

-The discussion section still requires improvement—strengthen the interpretation of findings, ensure logical flow, and link results to relevant literature.

Reviewer #3: The study conducted in Kenya on hepatitis B is important and warranted. However it lacks critical information needed to make conclusions. Below are some information required:

1. A flow diagram would be useful to assist with final number of specimens for analysis (denominator)

2. Demographic information of participants is lacking and important

3. Serological testing requires further information and expansion (interpretation of positive and negative results, errors, inconclusive results)

4. Line 134-138 requires revision and clarity for better understanding

4. Data analysis should be separated into bioinformatic analysis and statistical analysis (software used for calculations)

5. line 227-228 requires revision and very general and vague

6. line 238 needs revision and analyzed data

**Do you want your identity to be public for this peer review?** For information about this choice, including consent withdrawal, please see our Privacy Policy

Reviewer #1: No

Reviewer #2: No

Reviewer #3: **Yes:** Lesibana Malinga

---

## [Author Response · Author response to Decision Letter 1]

29 Dec 2025

Reviewer #1

Abstract

The authors should proofread the abstract; it seems AI was used to generate the abstract, hence there are many typos which need to be worked on - The authors have slightly reviewed the abstract for typos and also wish state that at no time in the writing of this manuscript was AI used.

Introduction

1.Some sentences could be more concise, for example, “These genotypes have shown a propensity for intergenotypic recombination…” could be tightened for flow. - We concur and this has now been revised for better flow.

2.The citation style (e.g., “Makhoha et al., 2023 recently published...”) should be in past tense without “recently,” as that becomes outdated; also, remove first names in scientific writing. - We agree with the reviewer and have revised as guided.

3.The paragraph on study populations (inmates vs. donors) is informative but slightly repetitive, it reiterates content from the methods and could be condensed. - We are in agreement and have deleted a sentence from this paragraph without losing the intended meaning.

4.The final sentence (“in line with the global agenda…”) is effective but would benefit from a stronger closing statement emphasizing the novelty or expected contribution of this study. - We are in agreement and have revised the sentence for a stronger closing statement.

Methods

1.Writing and organization

a)Some sentences are too long or contain redundant information, e.g., lines 135–139 could be shortened or merged for clarity. - we have reviewed the sentence for brevity and clarity.

b)There are minor grammatical inconsistencies (“then sequences” “then sequenced”; “was then sequences by bidirectional…” “was then sequenced bidirectionally”). Suggestion: Edit for flow and consistency (use past tense uniformly, e.g., “was performed,” “was analyzed”). - Apologies, this has now been corrected.

2.Scientific details missing or vague

a)Primer sequences are not explicitly listed; the reference “(17)” should be clearly identified and verified. One would expect at least the forward and reverse primer sequences or accession numbers. - The reference (Chook et al., 2015) has now been indicated, these are universal primers commonly used in HBV molecular studies. The primer co-ordinates are also given.

b)The annealing temperatures and cycle numbers are provided, but the amplicon size (bp) is not stated. - the amplicon size has been added after the description of the second round PCR.

c)The sequencing service could specify location or facility (e.g., “sequenced at Kenyatta University molecular lab or outsourced to …”). - the sequencing facility has been identified in the last paragraph of this section.

d)It is unclear how quality control of sequences (trimming, contamination check) was performed before alignment. Suggestion: Add brief information on amplicon size, primer details, and QC filtering steps. - a brief description of this has been included in the revision in the data analysis section.

e)Although the use of MrBayes is correct, the model of nucleotide substitution (e.g., GTR+G+I) used in the Bayesian analysis is not mentioned. Tree validation metrics (posterior probability thresholds, burn-in value, number of independent runs) are missing. - this has now been added in the revision to include the substitution model, burn-in and number of runs.

f)There is no mention of how mixed genotypes or ambiguous bases were handled. Suggestion: Include these parameters to strengthen reproducibility and perceived analytical rigor. - this information has been added in the MrBayes parameters description.

The overall tone is informative, but the phrasing could be made more formal and concise.

Example revision:

“Approximately 4 mL of venous blood was collected into plain vacutainers from each participant” is preferred over “Approximately 4 ml of blood was collected from a selected vein of each individual into a plain vacutainer.” - we agree with this comment and have revised the sentence.

Results

1.Seroprevalence of HBV

a)The confidence interval (CI) or exact chi-square value, which would improve statistical transparency is not defined or included. The phrase “did not reach statistical significance (p=0.31)” is correct but should be contextualised through discussing sample size limitations or low statistical power. Table 1 is mentioned but not fully integrated into the text (no data summary within the text). - The chi-square value has been added, pertinent data summary of Table 1 is already included in the text.

b)The Results section omits any age, sex, or risk factor stratification, which could enrich epidemiological interpretation. The seroprevalence analysis is scientifically sound but statistically underdeveloped and somewhat descriptive. Including more analytical insight (e.g., odds ratios or subgroup trends) would strengthen the epidemiological message. - This was intentionally left out in the current manuscript as this was included in a previous manuscript whose focus was on the seroprevalence results.

2.HBV DNA Detection and Genotyping

a)The authors report that “several seronegative samples” were successfully amplified this intriguing observation is scientifically important but underexplained. It suggests occult HBV infection (OBI), which warrants at least a brief acknowledgement even in the Results section. - We appreciate this comment and have acknowledged this observation.

b)The genotype distribution table is referred to as Table 1, but it should be Table 2 to maintain consistency (Table 1 has already used for seroprevalence). - Apologies for this, the correction has now been made.

c)The authors might clarify how the 22 samples were selected; were these all HBsAg-positive, or did they include a random subset? This is key for interpretation. - We appreciate this comment and have explained the selection of the 22 samples in the revision.

3.Phylogenetic Analysis

a)The latter probability (91%) is acceptable but could be accompanied by a mention of tree topology metrics or branch support for other genotypes for completeness. - A new sentence has now been added to report the branch support for the other genotypes.

b)The statement “distinctly clustered away from other sub-genotypes” should be supported by describing whether this divergence indicates a new lineage, regional adaptation, or recombination. - Since this observation contrasted with the geno2pheno results, the possibilities mentioned were considered necessitating recombination analysis. This was outlined in the discussion section.

c)The recombination results are briefly stated without genomic detail (e.g., breakpoint positions, affected regions). - The breakpoint positions have now been stated in the revision, additionally, the sequenced region is identified as the P/S gene.

d)The analysis would be stronger if the authors indicated the potential epidemiological or clinical implications of these recombinants, e.g., whether such patterns are consistent with previous East African HBV diversity. - We are in agreement and have included this in the discussion section.

Discussion

The authors open by linking their study to the UN SDG 3.3 and WHO’s 2030 hepatitis elimination goals, a globally relevant framing that gives the discussion strong policy alignment. They effectively justify the study’s focus on underrepresented populations (inmates, Northern Kenya), demonstrating clear research need.

a)However, the paragraph reads slightly like an introduction rather than a discussion. While context is valuable, a more analytical transition (e.g., “Our findings contribute to this goal by…”) would make it flow better into interpretation. The Citations to WHO and SDG literature could be condensed to avoid redundancy. The contextual introduction is excellent in scope but could be streamlined for brevity and better transition into study interpretation. - We sincerely appreciate this and are in total agreement. We have now revised this paragraph to fit the context.

The following should be worked on :

b)There is little exploration of potential behavioural or environmental factors driving higher inmate prevalence (e.g., overcrowding, shared instruments, limited healthcare access). - We appreciate this comment and have included these factors with relevant citations in the revision.

c)The text is descriptive and could integrate a short epidemiological interpretation beyond numeric comparison. The discussion is well-documented and comparative, but under-discussed regarding underlying causes and implications for screening policy. - this observation is correct, however this was done to avoid an overlap with the seroprevalence paper.

d)The text could make clearer whether genotype D dominance was observed across both populations or more pronounced in a specific one (this distinction is important for epidemiologic inference). - we concur with the reviewer and have explained this in the revision.

e)The discussion would benefit from hypothesising about possible sources or routes of introduction, for instance, historical or cross-border movement from Somalia or Ethiopia. - We had included this by hypothesizing that it was “through migration or trade”

f)A sentence or two connecting genotype D’s clinical implications (e.g., treatment response to tenofovir or interferon) would enhance translational impact. - We agree with the reviewer that this is important and included a sentence on it in the revision.

g)This section is scientifically rigorous and insightful but could emphasise why genotype D matters biologically beyond its novelty. - This had been mentioned but the increased rate of escape mutations in genotype D has been added.

h)The recombination discussion (lines 290–310) becomes somewhat repetitive, restating similar concepts (migration, cross-border mixing, recombination potential). It could be condensed without losing meaning. - We agree with the reviewer and have greatly revised the recombination discussion for clarity.

i)The authors could explicitly indicate whether their D/A and D/E recombinants share breakpoint regions previously reported in Africa; this would strengthen the evolutionary argument. - We are in agreement with the reviewer and have cited several studies from Africa that have reported these recombinants in the revision.

j)(Lines 316–340), Public Health and Diagnostic Implications were explicitly discussed; however, the potential impact on therapeutic outcomes could be supported by referencing known genotype D-associated resistance patterns (e.g., lower interferon response). - We agree that this is an important point and have added two sentences to support the public health implications of genotype D.

k)The final recommendations could briefly mention capacity-building needs (infrastructure, partnerships) for genomic surveillance in Kenya. - We appreciate this comment and have revised the final recommendations accordingly.

Reviewer #2

1.The mention of genotype D in the title is ambiguous, as there are also reports on genotype A. Please clarify whether genotype A samples were excluded from the analysis or adjust the title accordingly. - Thank you for this comment, we regret this mistake. In a bid to report the unusual trend we obscured genotype A from the title. We have now revised the title accordingly.

Manuscript Text:

a)Review the manuscript for grammar, sentence structure, and scientific writing style. - We have endeavored to review this in the revision.

b)Correct issues related to sentence case, punctuation (commas, full stops, etc.), and overall readability. - We have checked on this and made a few revisions to improve readability.

c)Revise unclear or awkward sentences as previously suggested. - Identified awkward sentences have now been revised.

d)Remove redundant or misplaced information throughout the text to improve flow and conciseness. - we have endeavored to generally improve the flow of the text.

e)Include missing details on statistical analysis, including the frequency and distribution of results. -The missing details identified by reviewer #1 have been incorporated in the revision.

f)Avoid repetition of information, especially in the results and discussion sections. - We are in agreement and have removed the identified repetitions pointed out by reviewer #1.

g)Ensure all factual statements are properly referenced where applicable. - All efforts have been made to reference he factual statements.

h)Add missing details on reagents and materials, including manufacturer and source (e.g., ExoSAP-IT™ (ThermoFisher Scientific, CA, USA)). - The details of the PCR reagents have been added that were erroneously missed out.

i)The discussion section still requires improvement—strengthen the interpretation of findings, ensure logical flow, and link results to relevant literature. - Several improvements pointed out by reviewer #1 have been incorporated in the revised draft.

Reviewer #3

Below are some information required:

1.A flow diagram would be useful to assist with final number of specimens for analysis (denominator) - The total number of samples had been included, however the number of HBV DNA samples was missing and has now been included in the revision.

2.Demographic information of participants is lacking and important - The demographic information was intentionally left out as it was part of a previously published manuscript.

3.Serological testing requires further information and expansion (interpretation of positive and negative results, errors, inconclusive results) - This has not been included due to the simplicity of the test cassette and the fact that no inconclusive / invalid results were observed.

4.Line 134-138 requires revision and clarity for better understanding. - We agree with this comment and have revised the two sentences for clarity.

5.Data analysis should be separated into bioinformatic analysis and statistical analysis (software used for calculations) - Thank for this comment, the statistical analysis bit had been left out but has now been added.

6.line 227-228 requires revision and very general and vague - We are in agreement and have greatly revised this paragraph for conciseness.

7.line 238 needs revision and analyzed data - We have indicated the data in the next sentence and hope that this is sufficient.

---

## [Decision Letter · Decision Letter 1]

2 Feb 2026

Molecular characterization of hepatitis B virus genotypes A and D among inmates and blood donors in Northeastern Kenya

PONE-D-25-53563R1

Dear Dr. Gachara,

We’re pleased to inform you that your manuscript has been judged scientifically suitable for publication and will be formally accepted for publication once it meets all outstanding technical requirements.

Kind regards,

Maemu Petronella Gededzha, Ph.D

Academic Editor

PLOS One

Additional Editor Comments (optional):

The authors have addressed all the comments to the reviewer's satisfactions.

Reviewers' comments:

Reviewer's Responses to Questions

**Comments to the Author**

Reviewer #2: All comments have been addressed

Reviewer #3: All comments have been addressed

2. Is the manuscript technically sound, and do the data support the conclusions?

Reviewer #2: Yes

Reviewer #3: Yes

3. Has the statistical analysis been performed appropriately and rigorously?

Reviewer #2: Yes

Reviewer #3: Yes

4. Have the authors made all data underlying the findings in their manuscript fully available?

Reviewer #2: Yes

Reviewer #3: Yes

5. Is the manuscript presented in an intelligible fashion and written in standard English?

Reviewer #2: Yes

Reviewer #3: Yes

Reviewer #2: The author has satisfactorily addressed the majority of the comments raised in the previous review. The revised manuscript shows clear improvement in grammar, sentence structure, and overall scientific writing style. Redundant and repetitive statements have largely been removed, particularly within the Results and Discussion sections, which has improved readability and flow.

Regarding the concern about the title’s emphasis on genotype D despite the inclusion of genotype A data, the author has now clarified this issue and revised the title accordingly. This change more accurately reflects the scope of the study and resolves the earlier ambiguity.

The manuscript has also been carefully checked for sentence case, punctuation, and formatting inconsistencies, and previously identified awkward or unclear sentences have been appropriately revised. Additional information on statistical analyses, including frequency distributions and reporting of results, has been incorporated as requested, addressing gaps previously highlighted. Furthermore, factual statements are now appropriately referenced, and missing methodological details have been added.

Overall, the revisions have strengthened the clarity, rigor, and transparency of the manuscript. The reviewer is satisfied that the major concerns have been addressed and considers the manuscript suitable for further consideration, subject to any final editorial checks that may be required

Reviewer #3: None at this stage all the comments have been met and revision made to the manuscript. Adequate data have been included for better understanding and clarity.

**Do you want your identity to be public for this peer review?** For information about this choice, including consent withdrawal, please see our Privacy Policy

Reviewer #2: No

Reviewer #3: **Yes:** Lesibana Malinga

---

## [Editor Report · Acceptance letter]

PONE-D-25-53563R1

PLOS One

Dear Dr. Gachara,

I'm pleased to inform you that your manuscript has been deemed suitable for publication in PLOS One. Congratulations! Your manuscript is now being handed over to our production team.

Kind regards,

on behalf of

Dr. Maemu Petronella Gededzha

Academic Editor

PLOS One